# Post-Harvest LED Light Irradiation Affects Firmness, Bioactive Substances, and Amino Acid Compositions in Chili Pepper (*Capsicum annum* L.)

**DOI:** 10.3390/foods11172712

**Published:** 2022-09-05

**Authors:** Chaochao Liu, Hongjian Wan, Youxin Yang, Qingjing Ye, Guozhi Zhou, Xiaorong Wang, Golam Jalal Ahammed, Yuan Cheng

**Affiliations:** 1College of Biotechnology, Jiangsu University of Science and Technology, Zhenjiang 212018, China; 2State Key Laboratory for Managing Biotic and Chemical Threats to the Quality and Safety of Agro-Products, Institute of Vegetables, Zhejiang Academy of Agricultural Sciences, Hangzhou 310021, China; 3Jiangxi Key Laboratory for Postharvest Technology and Nondestructive Testing of Fruits &Vegetables, Collaborative Innovation Center of Post-Harvest Key Technology and Quality Safety of Fruits and Vegetables, College of Agronomy, Jiangxi Agricultural University, Nanchang 330045, China; 4Institute of Manage Science and Engineering, College of Hebei Architecture and Civil Engineering, Zhangjiakou 075132, China; 5College of Horticulture and Plant Protection, Henan University of Science and Technology, Luoyang 471023, China; 6Henan International Joint Laboratory of Stress Resistance Regulation and Safe Production of Protected Vegetables, Luoyang 471023, China

**Keywords:** pepper, LED irradiation, post-harvest quality, capsaicin, amino acid composition

## Abstract

Chili pepper is an important vegetable and spice crop with high post-harvest deteriorations in terms of commercial and nutritional quality. Light-emitting diodes (LEDs) are eco-friendly light sources with various light spectra that have been demonstrated to improve the shelf-life of various vegetables by manipulating light quality; however, little is known about their effects on the post-harvest nutritional quality of chili peppers. This study investigated the effects of different LED lightings on the post-harvest firmness and nutritional quality of chili peppers. We found that red and blue light could increase the content of capsaicinoids, whereas white and red light could increase the essential and aromatic amino acid (AA) content in pepper. Nonetheless, the influence of light treatments on AA contents and compositions depends strongly on the pepper genotype, which was reflected by total AA content, single AA content, essential AA ratio, delicious AA ratio, etc., that change under different light treatments. Additionally, light affected fruit firmness and the content of nutrients such as chlorophyll, vitamin C, and total carotenoids, to varying degrees, depending on pepper genotypes. Thus, our findings indicate that LED-light irradiation is an efficient and promising strategy for preserving or improving the post-harvest commercial and nutritional quality of pepper fruit.

## 1. Introduction

Chili pepper (*Capsicum annum* L.) is the most widely cultivated condiment and vegetable crop worldwide [1,2]. The widespread consumption of chili pepper is attributed to its unique spicy characteristic, conferred by capsaicinoids. Capsaicin and dihydrocapsaicin constitute more than 80% of capsaicinoids, which mainly determine the pungency of pepper [2]. In addition to the pungency flavor, capsaicinoids have also been demonstrated to exhibit antioxidant, anticarcinogenic, anti-inflammatory, and thermogenic properties [3]. Moreover, fresh peppers are also rich in amino acids and bioactive substances, including chlorophyll, vitamin C, carotenoids, etc., which can effectively scavenge active oxygen free radicals in human body and reduce the risk of cardiovascular and cerebrovascular diseases and cancer [3,4,5]. However, post-harvested fresh peppers maintain vigorous metabolic activities due to their high moisture content, resulting in reduced biologically active substances, thereby reducing their nutritional and health benefits. Therefore, developing methods to prolong the shelf life of harvested peppers and maintain their nutritional quality has attracted extensive attention from researchers.

Light-emitting diodes (LEDs), which are usually used to manipulate the light environment, have been applied as an environmentally friendly means to improve plant tolerance to diverse stresses, or prolong the shelf-life and maintain the post-harvest quality of horticultural crop products over the last decades [6,7,8]. For instance, red or blue light irradiation has been proven to be able to increase the accumulation of AsA, total phenolics, and total sugars in post-harvested tomato, banana, Chinese bayberry, and strawberry fruit [9,10,11,12]. Although many studies have demonstrated the positive effects of LED irradiation on the post-harvest and nutritional qualities of vegetables and fruit, these positive effects depend on various factors, such as crop species, genotype, light quality, and irradiation duration. For example, LED light irradiation delayed the chlorophyll degradation process of purple kale and pak choi, whereas it accelerated banana de-greening during post-harvest storage [9,13,14].

To date, only a few studies concerning the effect of LED light on the post-harvest quality of pepper fruit had been reported and they were conducted with limited kinds of LED lightings, pepper genotypes, and quality-related parameters [15,16]. Therefore, in order to comprehensively assess the implementation potential of LED technology on post-harvested peppers, we investigated the effects of three different LED lightings (red, blue, and white) on the fruit qualities of three pepper genotypes. The findings obtained from this study would further improve our understanding of LED lighting application in post-harvested pepper storage.

## 2. Materials and Methods

### 2.1. Plant Materials and Growth Conditions

Three hybrid cultivars of pepper, ‘Hangjiao-2’, ‘Xinxiang-2’, and ‘P1622’, each displaying a distinct phenotype (Table 1), were used in this study. We planted 20 seedlings for each cultivar in the Yangdu Experimental Field at the Zhejiang Academy of Agricultural Sciences, Zhejiang Province (120°2′ E, 30°27′ N) in April 2019 and let them grow until July 2019. Thereafter, 48 pepper fruits were collected from each variety at the mature green stage (35 d after pollination) and subjected to different light irradiation treatments. The fruit was then divided into four groups (containing 12 fruits each), among which three groups were placed under continuous LED white light (50 μmol·m^−2^·s^−1^, spectrum is shown in Appendix A), LED red light (660 nm, 50 μmol·m^−2^·s^−1^), or LED blue light (450 nm, 50 μmol·m^−2^·s^−1^), for 48 h and the remaining group was placed in the dark as a control. Three biological replicates, each consisting of 4 fruits, were set for each treatment, with a temperature of 24 °C and relative humidity of 75 ± 3%. After light treatment, fruits were randomly collected from each biological replicate, and their firmness was measured using a fruit pressure tester (Model-GY3, Tuopu, Hangzhou, China). Subsequently, the pericarps of fruit were sampled and immediately frozen in liquid nitrogen and stored at −80 °C for further analysis.

### 2.2. Soluble Protein Content Determination

The soluble protein content was determined based on a previously reported method [17] but with slight modifications. Briefly, 0.3 g of frozen powdered sample was used for protein extraction with 3 mL of extraction buffer (200 mM Tris, 10 mM EDTA, pH 5.5). The protein content of the obtained supernatant was then detected using Coomassie brilliant blue G-250 (Shenggong, Shanghai, China) at the UV absorbance value of 595 nm. Thereafter, bovine serum albumin (Sigma-Aldrich, Shanghai, China) was used as the standard to calculate the soluble protein content, expressed as g kg^−1^ of the fresh weight.

### 2.3. Determination of Total Chlorophyll and Carotenoids Contents

The extraction and determination of total chlorophyll and carotenoid contents were conducted according to Zhao et al. [17]. Briefly, 3 mL of acetone was added to the homogenized sample for ultrasonic extraction, after which the total chlorophyll and carotenoid contents were separated by a C18 liquid phase column (4 μm, 3.9 mm × 150 mm, Elite, Dalian, China) of a Waters 600 HPLC system. The separated chlorophyll and carotenoids were then detected by a UV spectrophotometer at 428 nm and 448 nm, respectively. The content of chlorophyll and carotenoids were expressed as mg kg^−1^ of the fresh weight.

### 2.4. HPLC Analysis of Capsaicin and Ascorbic Acid (Vitamin C)

Capsaicin was extracted and quantified using a modified procedure from Arce-Rodriguez and Ochoa-Alejo [18]. Briefly, 10 g of frozen fruit pericarps of each sample was ground into powder and lyophilized for 48 h. Thereafter, the samples were extracted with 5 mL of methanol and incubated in a water bath at 60 °C for 30 min. The extract was filtered twice using filter papers, centrifuged at 14,000× *g* for 15 min, and the supernatant was passed through a 0.45 μm filter membrane (Shenggong, Shanghai, China). Capsaicin and dihydrocapsaicin were separated on an HPLC system (Waters Corp., Milford, MA, USA) using a Waters 600 separation module and 2414 RI detector, with methanol/water (65/35, *v*/*v*) as the mobile phase. The flow rate was 1 mL min^−1^, and the ultraviolet detection wavelength was 280 nm. Each sample was measured in 3 biological replicates, and the concentration of capsaicinoids (including capsaicin and dihydrocapsaicin) was expressed as mg kg^−1^ of fresh weight.

Vitamin C was extracted according to the modified methods of Liu et al. [19]. The freeze-dried fruit pericarps were ground into powder, and 0.2 g of each sample was extracted with an oxalic acid solution (1%, *w*/*v*). Thereafter, the extracts were centrifuged, and supernatants were collected for further HPLC analysis. Samples were injected into an HPLC system and separated in a C18 column (5 μm, 4.6 mm × 250 mm, Elite instrumental Ltd., Dalian, China) at a flow rate of 1 mL min^−1^ using 0.1% oxalic acid (*w*/*v*) as mobile phase. Vitamin C content was calculated at the absorbance value of 243 nm using an authentic ascorbic acid (Sigma-Aldrich, Shanghai, China) as the standard and expressed as mg kg^−1^ of fresh weight.

### 2.5. Total Amino Acid Composition Determination

Fruit homogenate (100 mg) was placed in a 20 mL hydrolysis tube, and 10 mL of 6 M HCl was added. The tubes were then evacuated using a vacuum pump, filled with high-purity nitrogen, and sealed. Subsequently, the sealed samples were hydrolyzed at 110 °C for 24 h and dried in a vacuum desiccator at 55 °C. The dried samples were then dissolved in 5 mL of sodium citrate buffer (pH 2.2), and then the supernatants were filtered through 0.22 μm membrane filters. The amino acid content was determined with a Hitachi L8800 amino acid analyzer (Hitachi High-Technologies Co., Tokyo, Japan) equipped with a 262 0 MSC-PS column (80 × 4.6 mm). Samples were analyzed by comparing peak profiles of the obtained samples with standard amino acid profiles. The content of amino acids was expressed as g kg^−1^ of fresh weight in Table 2 and Appendix A. As for amino acid nutrition valuation in Table 3, the percentage of each human essential amino acid to total amino acids was calculated and compared with the amino acid pattern spectrum reported by FAO/WHO/UNU [20].

### 2.6. Statistical Analysis and Design

A completely randomized design with at least three replicates, each comprising 4 fruits, was used. Statistical analysis of the bioassays was performed with SPSS statistical software (ver.19.0, SPSS Inc., Chicago, IL, USA), using a one-way analysis of variance. Shared letters indicate no statistically significant difference in the means (*p* > 0.05) after using Tukey’s test.

## 3. Results

### 3.1. Effects of Light Treatments on Capsaicinoid Content of Pepper Fruits

Different light qualities had significant effects on the content of capsaicinoids (capsaicin and dihydrocapsaicin) in the pepper fruits (Figure 1). The capsaicinoid content of all three cultivars increased under red and blue light compared with the control. However, the increase in both capsaicin and dihydrocapsaicin in ‘Xinxiang-2’ was not as high as in the other two cultivars under red and blue light treatments (Figure 1A,B). For instance, the capsaicin of ‘Xinxiang-2’ increased by 44% and 45% under red and blue light, respectively, whereas it was 53% and 48% in ‘Hangjiao-2’, and notably, 88% and 80% in ‘P1622’. As for white light treatment, capsaicin levels of ‘Xinxiang-2’ and ‘P1622’ were slightly increased by about 20% compared with the control, whereas the dihydrocapsaicin content was not affected. In contrast, the white light reduced both the capsaicin and dihydrocapsaicin contents in the cultivar ‘Hangjiao-2’ by approximately 20% (Figure 1A,B). As a result, all light treatments (white, red, and blue) induced total capsaicinoid (capsaicin and dihydrocapsaicin) accumulation in the three tested cultivars, except for the white light treatment on ‘Hangjiao-2’ (Figure 1C). Furthermore, as demonstrated in Figure 1A,B, the extent of increase of capsaicin was always higher than that of dihydrocapsaicin in ‘Xinxiang-2’ and ‘P1622’, hence the capsaicin accumulation seemed to be more light sensitive compared with dihydrocapsaicin in these two cultivars.

### 3.2. Effect of Light Treatments on the Firmness of Pepper Fruits

Fruit firmness refers to the pulp strength against pressure and is an important indicator of fruit quality, affecting the shelf life, flavor, and taste [21]. The white-light irradiation decreased the fruit firmness of all three pepper varieties (Figure 2), with that of the cultivar ‘Xinxiang-2’ being reduced by as much as 56.9%. Red- and blue-light treatments also led to a certain level of firmness drop in the ‘Xinxiang-2’ fruit, by 19.8% and 14.7%, respectively. In contrast, the red-light treatment enhanced the firmness of the ‘Hangjiao-2’ fruit, and the blue-light treatment increased the firmness of the ‘P1622’ fruit. The results of firmness change indicate that the effect of light on fruit firmness might depend on the genotype of the pepper.

### 3.3. Effects of Light Treatments on the Nutritional Quality of Pepper Fruits

Chlorophyll, carotenoids, vitamin C (Vc), and soluble protein are important nutritional value indicators in pepper. Different light treatments decreased the chlorophyll content of ‘Xinxiang-2’ and ‘P1622’ (except for white light treatment of ‘P1622’), but had no significant effect on ‘Hangjiao-2’ (Figure 3A). As for the total carotenoid content, increases were observed in ‘Hangjiao-2’ under both white light by 18% and blue light by 28%, and also ‘Xinxiang-2’ under red light by 25%, whereas a decrease of 14% was observed in ‘Xinxiang-2’ under blue-light treatment. Notably, all light treatments had no significant effect on the total carotenoid accumulation of the ‘P1622’ cultivar (Figure 3B). The results of Vc content showed that only white light irradiation increased the Vc content of ‘Hangjiao-2’ compared with the control, and the red light had no effect on Vc accumulation, whereas the blue light reduced the Vc content in all three cultivars (Figure 3C). According to Figure 3D, the total soluble protein content of ‘Hangjiao-2’ and ‘Xinxiang-2’ decreased under all three light treatments. However, both white and blue light increased the soluble protein content of ‘P1622’ (Figure 3D). The above results show that, similar to fruit firmness, the effects of light quality on the post-harvested fruit nutrients were also dependent on the genotype of pepper varieties.

### 3.4. Effects of Light Treatments on the Amino Acid Composition and Content of Pepper Fruits

Amino acid content affects the nutritional quality and taste of vegetables. We here identified 17 amino acids (AAs), including 7 essential and 10 non-essential ones from the pepper fruits (2 AAs essential for children included). Among the total 17 AA identified, 8 were delicious amino acids (DAAs), including 2 monosodium-glutamate-like AAs, 4 sweet AAs, and 2 aromatic AAs (Appendix A), which could give the pepper fruit umami, sweet, and aroma flavors, respectively. Under all light treatments, the contents of most amino acids in ‘Hangjiao-2’ were higher than that of the control (dark), with red and blue light having a more remarkable effect than white light. Conversely, light treatments decreased the content of most amino acids in ‘P1622’, with the most remarkable decrease under white light irradiation. Notably, both white and red light reduced the content of several amino acids, such as Asp, Met, and Arg, in ‘Xinxiang-2’, whereas blue light increased the content of these amino acids. Overall, different light qualities could have differential effects on the amino acid composition and contents of pepper fruits.

We further analyzed the content and composition ratio of different amino acid categories. We found that the effects of the three light qualities on different amino acid categories are similar to their effects on individual amino acids. Specifically, white light increased the ratio of the essential amino acid (A) and aromatic amino acid (F), whereas red light increased essential amino acid (A) and sweet amino acid ratios (E) across the three pepper varieties. Conversely, the white light decreased the ratio of the non-essential amino acid (B), children essential amino acid (C), and MSG-like amino acid (D), whereas the red light reduced the proportion of non-essential amino acid (B) across the three cultivars. The blue light increased essential amino acid (A) and MSG-like amino acid (D), and decreased non-essential amino acid (B) in ‘Hangjiao-2’, whereas it only decreased sweet amino acid (E) in ‘P1622’ (Table 2).

**Table 2 foods-11-02712-t002:** Effect of light quality on the amino acid composition and content (g kg^−1^) in fruits of three pepper cultivars.

	Hangjiao-2				Xinxiang-2				P1622			
	Dark	White	Red	Blue	Dark	White	Red	Blue	Dark	White	Red	Blue
T	37.0 ±2.1 c	40.6 ±2.3 b	54.3 ±3.6 a	52.1 ±3.1 a	37.2 ±2.1 b	35.0 ±2.0 c	33.8 ±1.9 c	38.8 ±2.4 a	36.4 ±1.9 a	31.6 ±1.6 b	32.4 ±1.7 b	32.8 ±1.9 b
A	14.8 ±0.8 c	17.8 ±1.1 b	22.3 ±1.5 a	21.4 ±1.4 a	14.9 ±1.0 ab	14.7 ±0.9 ab	14.3 ±0.9 b	15.5 ±1.1 a	13.9 ±0.7 a	12.7 ±0.6 b	13.0 ±0.6 b	12.6 ±0.6 b
B	22.3 ±1.3 b	22.8 ±1.2 b	31.9 ±2.1 a	30.8 ±1.7 a	22.3 ±1.2 a	20.3 ±1.1 b	19.5 ±1.1 b	23.3 ±1.3 a	22.5 ±1.2 a	18.9 ±1.0 c	19.4 ±1.1 c	20.2 ±1.3 b
C	3.1 ±0.2 b	3.3 ±0.2 b	4.7 ±0.3 a	4.5 ±0.4 a	3.3 ±0.1 b	3.0 ±0.2 c	2.7 ±0.1 c	3.6 ±0.2 a	3.8 ±0.3 a	3.0 ±0.2 c	2.8 ±0.2 d	3.6 ±0.2 b
D	7.1 ±0.5 b	6.8 ±0.3 b	11.1 ±0.8 a	10.8 ±0.5 a	8.0 ±0.5 a	6.6 ±0.3 b	6.0 ±0.3 c	8.2 ±0.5 a	8.0 ±0.4 a	6.4 ±0.3 d	6.7 ±0.3 c	7.2 ±0.4 b
E	6.9 ±0.5 c	8.1 ±0.6 b	10.5 ±1.0 a	9.9 ±0.9 a	7.0 ±0.4 a	6.6 ±0.5 b	6.5 ±0.6 b	7.3 ±0.6 a	6.9 ±0.4 a	6.0 ±0.4 b	6.3 ±0.5 b	6.0 ±0.5 b
F	4.0 ±0.2 c	5.0 ±0.3 b	5.7 ±0.4 a	5.6 ±0.3 a	4.0 ±0.3 ab	4.1 ±0.2 ab	3.9 ±0.3 b	4.2 ±0.3 a	3.7 ±0.2 a	3.5 ±0.2 b	3.6 ±0.2 ab	3.4 ±0.2 b
A/T%	39.84 ±2.2 c	43.78 ±2.7 a	41.14 ±2.8 b	40.96 ±2.7 b	39.97 ±2.7 b	41.97 ±2.6 a	42.26 ±2.7 a	39.90 ±2.8 b	38.30 ±1.9 b	40.07 ±1.9 a	40.20 ±1.9 a	38.51 ±1.8 b
B/T%	60.16 ±3.5 a	56.22 ±3.0 c	58.86 ±3.9 b	59.04 ±3.3 b	60.03 ±3.2 a	58.03 ±3.1 b	57.74 ±3.3 b	60.10 ±3.4 a	61.70 ±3.3 a	59.93 ±3.2 b	59.80 ±3.4 b	61.49 ±4.0 a
C/T%	8.50 ±0.5 a	8.04 ±0.5 b	8.66 ±0.6 a	8.66 ±0.8 a	8.99 ±0.3 a	8.44 ±0.6 b	8.09 ±0.3 c	9.17 ±0.5 a	10.36 ±0.8 ab	9.56 ±0.6 b	8.54 ±0.6 c	10.81 ±0.6 a
D/T%	19.27 ±1.4 b	16.68 ±0.7 c	20.45 ±1.5 a	20.73 ±1.0 a	21.50 ±1.3 a	18.79 ±0.9 b	17.75 ±0.9 c	21.21 ±1.3 a	21.95 ±1.1 a	20.16 ±0.9 b	20.81 ±0.9 b	21.94 ±1.2 a
E/T%	18.68 ±1.4 b	19.89 ±1.5 a	19.43 ±1.8 a	18.93 ±1.7 b	18.86 ±1.1 b	18.64 ±1.4 b	19.20 ±1.8 a	18.78 ±1.5 b	18.99 ±1.1 b	19.02 ±1.3 b	19.41 ±1.5 a	18.18 ±1.5 c
F/T%	10.72 ±0.5 b	12.26 ±0.7 a	10.54 ±0.7 b	10.83 ±0.6 b	10.75 ±0.8 b	11.63 ±0.6 a	11.64 ±0.9 a	10.81 ±0.8 b	10.10 ±0.5 b	11.00 ±0.6 a	10.99 ±0.6 a	10.44 ±0.6 ab

Note: T, Total amino acid; A, Essential amino acid; B, Non-essential amino acid; C, Children essential amino acid; D, Monosodium glutamate-like amino acid; E, Sweet amino acid; F, Aromatic amino acid. The results are shown as the mean ± SE of triplicate samples. Means denoted by the same letter did not differ significantly at *p* < 0.05 according to Tukey’s test.

### 3.5. Effect of Light Treatments on the Nutritional Value of Amino Acids

The amino acid pattern spectrum has been widely used to evaluate protein quality for human nutrition since being reported by FAO/WHO/UNU in 1970, and was updated in the years 1985 and 2007 [20]. We adopted the pattern to analyze the effect of light qualities on the amino acid nutrition value of pepper fruits. The results show that in the fruit of control and different light quality treatments using the three cultivars tested, most essential amino acids except for Met + Cys, meet the requirement for recommended amino acid pattern, indicating that Met + Cys was the first limiting amino acid in our experimental pepper varieties. In ‘Xinxiang-2’ and ‘P1622’, the red-light treatment had better effects on most essential amino acid accumulations compared with the control and other light qualities, whereas the blue light demonstrated a contrary tendency (Table 3). In ‘Hangjiao-2’, the white light had better-promoting effects on most essential amino acids, including Val, Ile, Leu, Lys and Phe + Tyr compared with the control and other treatments (Table 3). Notably, the results show that the white light increased the Leu and Phe + Tyr pattern whereas the red light increased Thr across the three cultivars.

**Table 3 foods-11-02712-t003:** Effect of light quality on the essential amino acid score of three pepper cultivars.

Essential Amino Acids	Amino Acid Scoring Pattern *	Hangjiao-2				Xinxiang-2				P1622			
		Dark	White	Red	Blue	Dark	White	Red	Blue	Dark	White	Red	Blue
Thr	2.30	5.34 ±0.27 a	5.29 ±0.25 a	5.50 ±0.37 a	5.35 ±0.19 a	5.20 ±0.27 b	5.31 ±0.29 b	5.55 ±0.30 a	5.19 ±0.26 b	5.09 ±0.27 b	5.04 ±0.32 b	5.33 ±0.31 a	4.90 ±0.61 c
Val	3.90	5.68 ±0.27 b	5.92 ±0.25 a	5.36 ±0.37 c	5.42 ±0.38 c	5.29 ±0.27 ab	5.48 ±0.29 a	5.38 ±0.30 ab	5.12 ±0.26 b	4.96 ±0.27 b	5.22 ±0.32 a	5.37 ±0.31 a	5.05 ±0.30 b
Ile	3.00	4.28 ±0.27 a	4.39 ±0.25 a	4.23 ±0.37 a	4.29 ±0.19 a	4.13 ±0.27 b	4.45 ±0.29 a	4.44 ±0.30 a	4.07 ±0.52 b	3.82 ±0.27 b	4.18 ±0.32 a	4.16 ±0.31 a	4.23 ±0.30 a
Leu	5.90	10.36 ±0.54 b	11.31 ±0.99 a	10.44 ±0.55 b	10.54 ±0.77 b	10.26 ±0.81 b	10.87 ±0.86 a	11.12 ±0.59 a	10.29 ±0.77 b	9.80 ±0.55 b	10.43 ±0.32 a	10.19 ±0.31 ab	9.84 ±0.30 b
Lys	4.50	7.72 ±0.54 b	8.34 ±0.49 a	7.77 ±0.55 b	7.86 ±0.58 b	7.46 ±0.54 ab	7.71 ±0.57 a	7.25 ±0.30 b	7.50 ±0.26 ab	7.20 ±0.27 b	7.46 ±0.32 ab	7.61 ±0.31 a	7.25 ±0.30 b
Met + Cys	2.20	1.78 ±0.11 b	1.66 ±0.12 b	1.96 ±0.13 a	1.57 ±0.10 b	1.79 ±0.08 a	1.86 ±0.11 a	1.89 ±0.09 a	1.89 ±0.08 a	1.98 ±0.11 a	1.79 ±0.13 b	1.60 ±0.09 c	1.61 ±0.09 c
Phe + Tyr	3.80	11.09 ±0.54 b	12.26 ±0.74 a	10.54 ±0.74 c	10.83 ±0.58 bc	10.75 ±0.81 b	11.63 ±0.57 a	11.64 ±0.89 a	10.81 ±0.77 b	10.10 ±0.55 c	11.00 ±0.63 a	10.99 ±0.62 a	10.44 ±0.61 b

*, Amino acid composition of an “ideal” protein as published by FAO/WHO/UNU [20]; Amino acids are represented by the 3-letter abbreviation code. The results are shown as the mean ± SE of triplicate samples. Means denoted by the same letter did not differ significantly at *p* < 0.05 according to Tukey’s test.

## 4. Discussion

Light plays a pivotal role in regulating the growth and development, environmental stress tolerance, and secondary metabolism of plants [22]. LED light sources have been widely used for post-harvest preservation and nutritional quality improvement of leafy vegetables and fruits [23]. This study evaluated the application of LED lighting to the post-harvest quality preservation of peppers.

Capsaicin and dihydrocapsaicin are a class of alkaloids belonging to secondary metabolites and are the two main pungent substances that determine the commercial quality of peppers, and their contents depend on genotypes and various environmental factors [2]. Gangadhar et al. [16] reported that blue LED light could promote the synthesis of capsaicin during fruit development, whereas red LED light had no significant effect compared with fluorescent light. According to Yap et al. [24], both supplemental red and blue light treatments for 5 h per day throughout the growing stage could increase the capsaicinoid content in chili fruits, with the blue light demonstrating a better promoting effect. Nevertheless, the effect of post-harvest irradiation with different light quality on the capsaicinoid content in peppers was not well studied. We found that red and blue light increased capsaicin and dihydrocapsaicin contents compared with dark or white light in all the tested pepper varieties, whereas white light could only slightly increase capsaicin content in ‘Xinxiang-2’ and ‘P1622’. This indicates that a single-band light has a more pronounced effect on the accumulation of capsaicinoid content in post-harvested chili peppers. Our results are not only in agreement with the previous studies conducted by Gangadhar et al. [16] and Yap et al. [24], but also further confirmed that red and blue LED lights can be used for pre- and post-harvest treatments to enhance capsaicinoid content, thereby increasing the pungency of different chili varieties. The more pronounced effects of red and blue light may be attributed to the fact that they are the two major types of light driving photosynthate biosynthesis, which is needed for the formation of secondary metabolites such as capsaicinoids.

Fruit firmness is an important commercial quality of chili peppers that is strongly associated with changes in water loss rate, cellular organelle integrity and cell wall composition [25,26,27,28]. We found that red and blue light maintained a greater firmness in ‘Hangjiao-2’ and ‘P1622’, whereas white light decreased fruit firmness in the three cultivars we studied. This is consistent with the findings of the study on LED-treated tomato fruits by Najera et al. [29], who reported that LED light of a higher red:far-red ratio positively affected the firmness, whereas white light negatively affected the fruit firmness of several tomato cultivars. Moreover, Dhakal and Baek [30] also reported that both red and blue light maintained post-harvest firmness of mature green tomato fruits three days after treatment and that blue light maintained its positive effect until the seventh day of the treatment. Thus, these results further demonstrate that the effect of light on fruit firmness during the post-harvest period depends on the light quality, crop genotype, and storage duration. However, to have a better understanding of the effect of light irradiation on postharvest peppers, more physiological indicators, such as stomatal opening, transpiration rate, and respiration rate, as well as the content of cell wall constituents (pectin, cellulose, and hemicellulose), and, in addition, gene expression level and enzymatic activities associated with these biological processes should be tested.

As for the nutritional quality change, both red and blue light irradiation decreased the content of chlorophyll in ‘Xinxiang-2’ and ‘P1622’. This is contrary to some studies that reported that red or blue light inhibited the degradation process of chlorophyll in some leafy vegetables, such as cabbage, purple kale, and pak choi leaves [13,14,31]. However, it has also been reported that light treatment increased the level of color parameter values (a* and b*) or ratio of a*/b* and ethylene production, indicating an accelerated post-harvest chlorophyll degradation and senescence of peach, banana, and tomato fruits [9,29,32]. Similarly, Perez-Ambrocio et al. [15] also reported that blue and UV-C light accelerated chlorophyll degradation at the later storage stages of habanero pepper (*Capsicum chinense*) fruits. Therefore, these studies suggest that the effects of light on post-harvest chlorophyll degradation in leafy vegetables and fruits vary greatly and can be contradictory. These findings indicate that the underlying mechanism related to chlorophyll biosynthesis and degradation process affected by different light quality in post-harvest leafy vegetables and fruits is interesting and warrants more future research.

Amino acid content and composition play an important role in the nutritional quality of food. Moreover, as an important primary metabolite, amino acids are closely linked to energy and carbohydrate metabolism, the carbon–nitrogen budget, and secondary metabolism [33,34]. We observed in ‘Hangjiao-2’ that most amino acids in light-irradiated peppers were higher than that of the control (dark). One reason for this phenomenon might be that pepper fruit under dark treatment encountered an energy shortage that would activate respiratory pathways that use amino acids as alternative substrates [33]. Phe and Val are the two original amino acids for the biosynthesis of capsaicinoids [2]; however, we found that light treatment decreased the contents of Phe and Val in ‘Hangjiao-2’ but increased them in ‘P1622’ (Appendix A). Meanwhile, light treatment increased capsaicin and dihydrocapsaicin both in ‘Hangjiao-2’ and ‘P1622’ (Figure 1), indicating that Phe and Val have opposite correlations with capsaicinoids in the two varieties. This is possibly due to the fact that Phe and Val are also involved in the biosynthesis of other secondary metabolites, for example, Phe is also the precursor of a large number of flavonoids, which are ubiquitous in the plant kingdom [33,34]. Hence, a metabolome analysis would provide a comprehensive understanding of this phenomenon. Overall, our study provided some referential evidence for the subsequent study of light–amino acid interaction in horticultural plants in the future.

## 5. Conclusions

Light quality affects the commercial and post-harvest nutritional quality of pepper fruit, and this effect is also dependent on the crop genotype. Generally, red light increases the capsaicinoid content and the ratio of essential and aromatic amino acids but reduces chlorophyll and total protein content in chili peppers. Conversely, blue light also increases capsaicinoid content but decreases the Vc content of chili peppers. White light increases the essential and aromatic amino acid ratio but decreases pepper fruit firmness. Thus, our study demonstrates that LED light irradiation is an efficient and promising strategy to preserve or improve the post-harvest quality of pepper fruit, especially red and blue light could be used by producers who are interested in enhancing or collecting capsaicinoids. Nevertheless, to draw the potential of this technology, more studies, such as the quantification of other nutritional compounds including capsanthin and flavonoids, and understanding the mechanism in collaboration with molecular plant science, are required.

## Figures and Tables

**Figure 1 foods-11-02712-f001:**
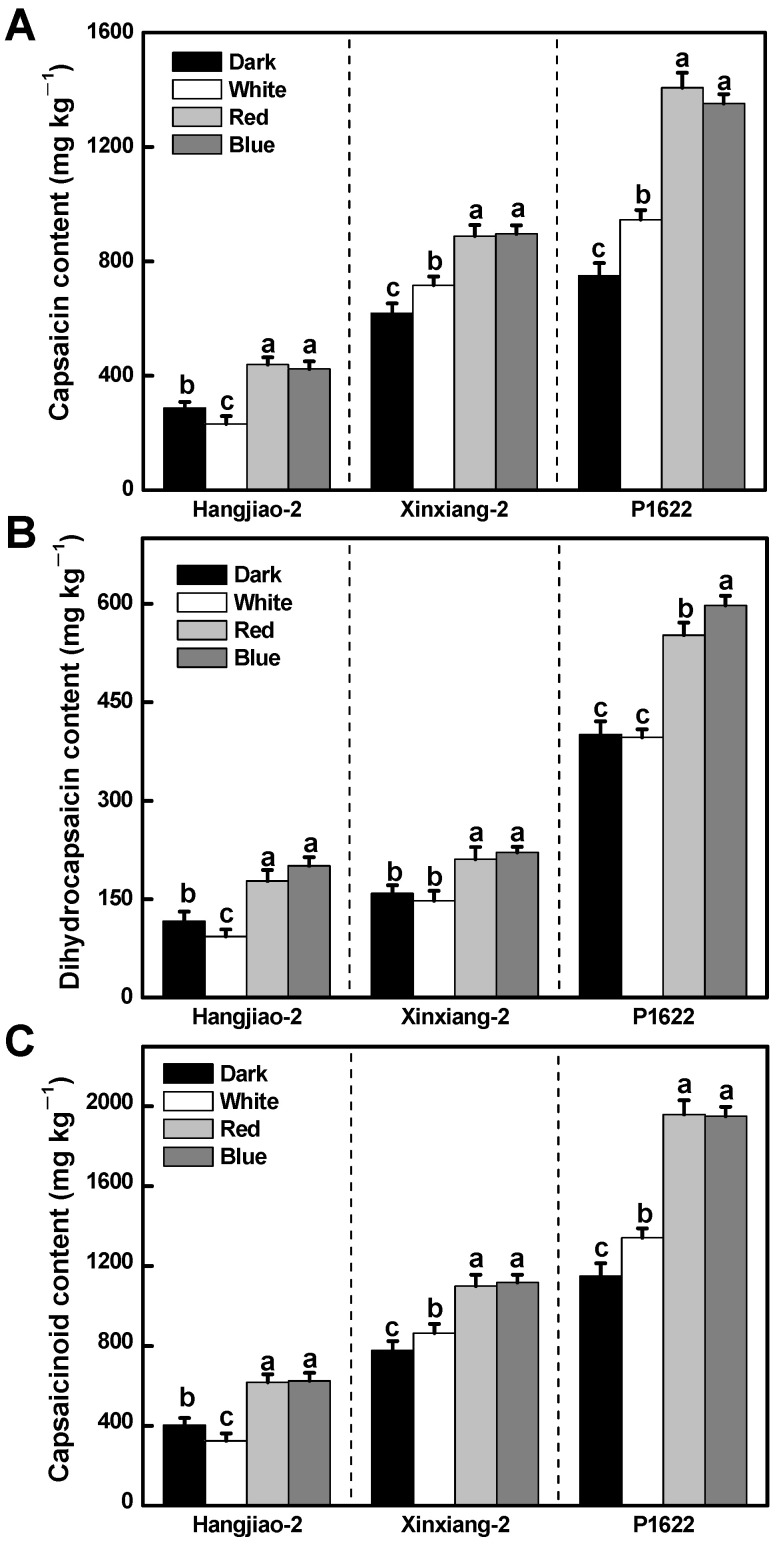
Effect of light quality on capsaicin (**A**), dihydrocapsaicin (**B**), and capsaicinoid (**C**) content in pepper fruits of three cultivars. Results are shown as mean ± SE of triplicate samples. Means denoted by the same letter were not significantly different at *p* < 0.05 according to Tukey’s test.

**Figure 2 foods-11-02712-f002:**
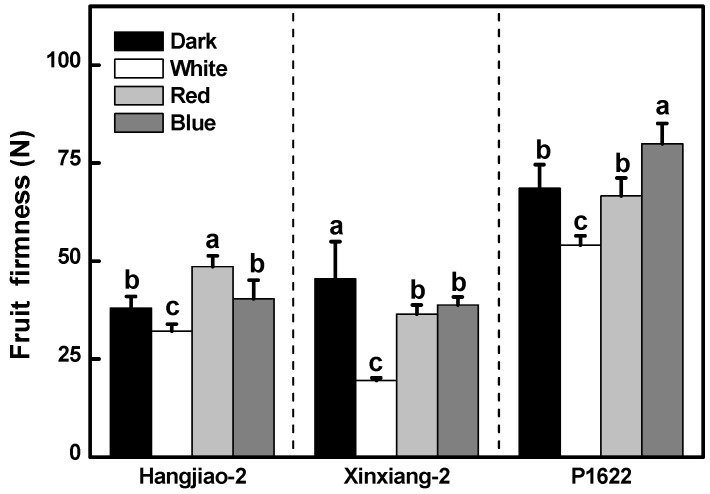
Effect of light quality on fruit firmness. Results are shown as mean ± SE of twelve samples. Means denoted by the same letter were not significantly different at *p* < 0.05 according to Tukey’s test.

**Figure 3 foods-11-02712-f003:**
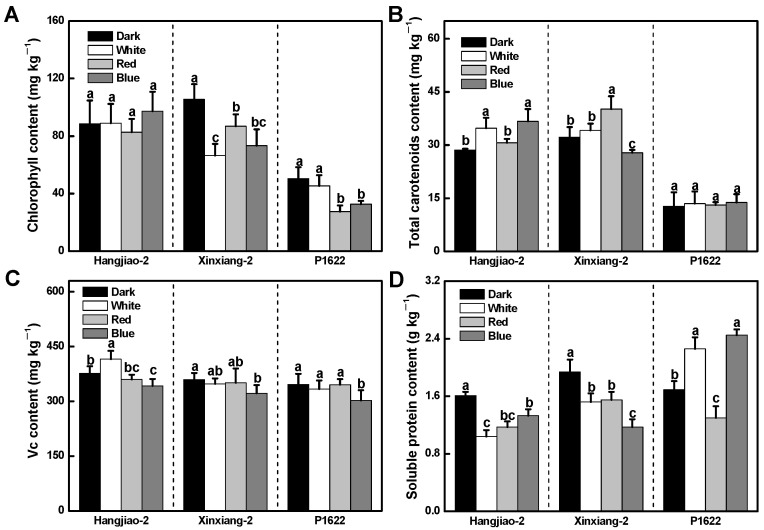
Effect of light quality on chlorophyll content (**A**), total carotenoid content (**B**), vitamin C content (**C**), and soluble protein content (**D**). Results are shown as mean ± SE of triplicate samples. Means denoted by the same letter were not significantly different at *p* < 0.05 according to Tukey’s test.

**Table 1 foods-11-02712-t001:** Phenotypic characteristics of three pepper cultivars used in this study.

Cultivar	Plant Height (cm)	Canopy Width (cm)	Fruit Length (mm)	Fruit Diameter (mm)	Fruit No. Plant^−1^	Single Fruit Weight (g)	Fruit Yield (g plant^−1^)
Hangjiao-2	56.3 ± 2.2 b	72.0 ± 4.6 b	119.6 ± 3.5 b	22.4 ± 1.1 a	55.5 ± 1.9 b	18.5 ± 1.2 b	1014.8 ± 27.9 a
Xinxiang-2	67.8 ± 2.5 a	76.3 ± 3.3 a	134.6 ± 3.2 a	21.2 ± 1.0 b	44.8 ± 5.1 a	20.8 ± 1.0 c	924.6 ± 163.6 a
P1622	44.8 ± 2.8 c	71.8 ± 1.7 b	88.9 ± 4.4 c	11.4 ± 0.2 c	106.5 ± 8.3 c	5.2 ± 0.6 a	559.3 ± 92.4 b

Note: The results are shown as the mean ± SE for quintuplicate samples. Means denoted by the same letter were not significantly different at *p* < 0.05 according to Tukey’s test.

## Data Availability

The data presented in this study are available on request from the corresponding author.

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
