# Peer review of "Post-Harvest LED Light Irradiation Affects Firmness, Bioactive Substances, and Amino Acid Compositions in Chili Pepper (*Capsicum annum* L.)"

_foods, 2022, doi:10.3390/foods11172712_

Round 1

Reviewer 1 Report

This paper presents some important results on the LED Light-induced fruit quality enhancement in chili pepper.

The authors presented some important aspects of the postharvest management of chili.

However, I have a few suggestions for improvement:

1. Introduction is too long. Please concise. Please mention how LED light is justified at the production/growers level.

2. Fig. 1 is unnecessary as there is no clear appearance.

3. Always use standard units and symbols as well as formatting. e.g. g plant-1 instead of g/plant.

4. In the methodology, there should be a description of statistical analysis and design.

5. results should be clear and comparative. It MUST be concise.

6. Discussion is written superficially. Please relate the results properly and just on the basis of relevant references. Also, explain the outstanding results.

7. The authors have not followed the instruction to the authors. They largely overlooked the reference style and formats. None of them are numbered and also the styles are different. The numbering is useless. Actually, they are arranged alphabetically. These are just meaningless.

8. I think the authors are careless in writing and they should revise it properly and importantly follow the instructions for authors.

Reviewer 2 Report

The aim and the novelty character of the research should be better marked.

Line 35 the scientific name of the plant should be inserted in italics.

More details on amino acid composition determination (subparagraph 2.5) should be given.

A subparagraph including correlations of the different results should be inserted.

Major considerations should be included in the discussion.

Limits, advantages, practical applications should be highlighted in conclusions.

Reviewer 3 Report

Review of the article: Post-harvest LED Light Irradiation Affects Firmness, Bioactive Substances, and Amino Acid Compositions in Chili Pepper (Capsicum annum L.).

The paper is quite well written and deals with the interesting topic of the effect of radiation on the post-harvest quality of pepper fruits. However, I have a few comments about the experiment. Firstly, 4 fruits in biological replication seems very low. Twelve fruits treated with a given light collected from 20 plants is not a sufficient test group. White light is not described sufficiently, whether it is warm or cold - please specify colour temperature or spectrum. It should be very clearly stated that control is darkness. In my opinion, the introduction and discussion are not sufficient. The introduction should focus on peppers and post-harvest processes in the fruit - information on leafy vegetables is redundant. In the discussion, the authors should, in addition to referring to other studies, try to explain the phenomena taking place - why this is happening, what we know and what we don't yet know. The conclusions chapter is not particularly useful for scientists because it describes what has been observed without answering the question as to why this is so, nor is it useful for producers because it does not answer the question as to what kind of treatment is ultimately needed for the peppers. Perhaps it would be worthwhile to select the characteristics that are important in assessing the quality of peppers and make recommendations for them. And exactly - is the amino acid content so important in peppers - rather, the vegetable is consumed as a side dish or condiment - in small quantities - so the amount of amino acids a person eats will not be significant either. On top of this, the table presenting the amino acid results is absolutely not reader-friendly. Perhaps it would be worth giving the results for the individual amino acids in the supplementary and leave the groups in the text? Another concern - the authors show that there is 1-2 g/kg protein and amino acids (total) 30-54 g/kg - isn't there an error here? There are very large numbers in Table 2 - you may want to express them in other units and not show the numbers after the decimal point - leave only the significant ones. From the description it is very difficult to see how the values in Table 3 were obtained - please complete the description. Why do the authors use Student's t-test to compare 4 means and not analysis of variance? On page 4 in the description of the amino acid determination the authors wrote "evacuated" and I think it should be evaporated? In chapter 3.3, figs 4c and 4d are mixed up. To sum up, the work seems interesting, but in its current form - it needs significant improvements.

 Author Response

Reviewer 4 Report

The authors have well executed the experiments and represented the results in a well mannered way.

The introduction part must be supported by latest review i.e. provided by latest references.

All the tables must be having the standard errors (S.E). Without SE all the values are meaningless.

Round 2

Reviewer 1 Report

This version is properly revised.

Author Response

We are grateful to reviewer 1 for your affirmation of our revised manuscript.

Reviewer 3 Report

The authors have made a few corrections. I understand that they cannot change the number of fruits in the repetition at the moment - but for the future 4 is very little. I understand the difference between soluble proteins and free amino acids, but I feel that the authors are misleading the reader. In the materials and methods section they write: that prior to the amino acid determination the sample was hydrolyzed in acid 110oC for 24 h - which would suggest that they are determining total amino acids, whereas they write that they are determining free amino acids. Authors cite the work of Gangadhar but comparing the amounts of proteins and amino acids in the two research there are huge differences - even differences of several tens of times. In my opinion, the amino acid content of the authors is overestimated - you can also compare with https://doi.org/10.1016/j.foodchem.2021.130797.

I disagree with response 2 - there is no equivalence between white LED = cold LED and in scientific research we should be precise and specific. Do the authors have any information on how much fresh or dried chili is eaten on average by a Chinese person per day? I have no idea. Only after such information we can wonder whether this amount contributes a significant amount of amino acids. Science is not based on faith (with reference to 'we believe').

The quality of figure 4 does not allow for analysis of these data and their description - it is very poor. In addition, the data shown in fig 4 do not relate to the title and aim of the work, i.e. to demonstrate the effect of radiation from LED lamps on the quality of peppers. This figure does not seem necessary.

Work still needs to be revised and clarified.
